# Establishment of a neonatal resuscitation registry in the Democratic Republic of the Congo: An open cohort study

Amy Mackay[1‡], Daniel Ishoso[2‡], Eric Mafuta[2], Joar Eilevstjønn[3], Patricia Gomez[4], Waldemar Carlo[1], Melissa Bauserman[5], Carl Bose[5], Jackie K. Patterson[5*]

1 Department of Pediatrics, University of Alabama at Birmingham School of Medicine, Birmingham, AL, 2 Kinshasa School of Public Health, University of Kinshasa, Kinshasa, Democratic Republic of the Congo, 3 Strategic Research, Laerdal Medical, Stavanger, Norway, 4 Jhpiego, Baltimore, MD, 5 Department of Pediatrics, University of North Carolina at Chapel Hill School of Medicine, Chapel Hill, NC

‡ AM and DI are co first-authors on this work.
* jackie_patterson@med.unc.edu

## Abstract

Improving neonatal resuscitation practices reduces neonatal mortality. In low- and middle-income countries (LMICs), granular details about provider actions during resuscitation are largely unknown; therefore, identifying targets for improvement is difficult. The International Liaison Committee on Resuscitation (ILCOR) recognizes the importance of uniform reporting of clinical neonatal resuscitation studies and published a guideline recommending specific variables to include. We established an open cohort study for newborn resuscitation in the Democratic Republic of the Congo (DRC) as a platform for developing and evaluating novel strategies to improve newborn resuscitation. We included all in-born neonates at two health facilities in Kinshasa, DRC. We gathered data on all enrollees via delivery registry and medical record abstraction. Using the Liveborn Observation app, we directly observed care at birth for a convenience sample. We collected heart rate data when providers used NeoBeat, a battery-operated heart rate meter. From September 2022 to August 2023, we abstracted delivery registry and medical record data for 6,414 newborns and gathered observational data on the infant's breathing status and provider actions for 3,166 (49%). Our dataset includes 85% of ILCOR's recommended core variables applicable to this setting, and 50% of ILCOR's applicable supplemental variables. Our registry also contains variables beyond those recommended by ILCOR that are contextually important for evaluating resuscitation care in LMICs such as duration of suctioning, pauses in positive pressure ventilation and fresh stillbirth. Our experience establishing a resuscitation registry with novel tools in the DRC serves as a model for resuscitation research in low-resource settings. Our cohort study provides important insight to inform subsequent versions of ILCOR's guideline on uniform reporting of neonatal resuscitations studies globally.

**Data availability statement:** All relevant data are within the paper and its supporting information files.

**Funding:** JP, Laerdal Foundation Program Award, https://laerdalfoundation.org/ The funder did not play any role in the study design, data collection and analysis, decision to publish or preparation of the manuscript.

**Competing interests:** The authors have declared that no competing interests exist.

## Introduction

The day of birth carries the highest risk of death during childhood [1]. Newborn resuscitation training reduces intrapartum-related deaths, but further reductions are limited by the necessary highly technical skills [2–4]. In the Democratic Republic of the Congo (DRC), we noted substantial gaps in care among providers trained in newborn resuscitation, including underuse, delays and interruptions in positive pressure ventilation (PPV) for newborns not breathing well by 60 seconds after birth [5].

Understanding what happens at the bedside during neonatal resuscitations is critical to improve resuscitation care. Currently, in low-resource settings, granular details about moment-to-moment provider actions during newborn resuscitation are rarely available. Furthermore, resuscitation studies in these settings often present baseline characteristics and neonatal outcomes (e.g., Apgars, neonatal mortality, hypoxic-ischemic encephalopathy) without the details of process indicators (i.e., provider actions) that contributed to these outcomes [6–8]. Understanding the timing, duration, and order of resuscitation practices is necessary to identify quality gaps and develop strategies to improve care.

The International Liaison Committee on Resuscitation (ILCOR) Neonatal Life Support Task Force recently published guidelines for uniform reporting of neonatal resuscitation in clinical studies [9]. The recommendations include core and supplemental variables within seven relevant domains: setting, patient, antepartum, birth/pre-resuscitation, resuscitation, post-resuscitation, and outcomes. Of note, the working group members all practice in highly resourced settings; to ensure the guideline was globally applicable, the working group solicited feedback from providers and investigators in low-resource settings and revised the guideline accordingly before finalization. As these guidelines are implemented globally, more in-depth engagement with a broad group of stakeholders in low-resource settings is needed to increase applicability to those settings [10,11]. ILCOR's recommendations are important for standardizing data elements globally to allow for better interpretation of individual studies and data synthesis across studies.

We established a resuscitation registry in health facilities in the DRC using novel tools, including a mobile health application for observational data collection of resuscitation events (the Liveborn Observation application [12]) and a battery-operated heart rate meter (NeoBeat). This registry is a platform to support development and evaluation of novel strategies to improve resuscitation care in low-resource settings. In this manuscript, we describe the tools used to establish our registry, the congruence of our dataset and ILCOR's recommended variables, and facilitators and barriers to resuscitation data collection in this low-resource setting.

## Materials and methods

In September 2022, we established a newborn resuscitation registry in two health facilities in Kinshasa, DRC, to collect granular data on clinical birth history, resuscitation care practices, and neonatal outcomes. In-born newborns, regardless of gestational age, weight, congenital anomalies, or vital status at birth (i.e., liveborn or stillborn) were eligible to participate in the registry.

## Description of health facilities

The two health facilities participating in this registry are operated by the Catholic church and located in urban Kinshasa, the capital of the DRC. Midwives and nurses are the primary providers responsible for early newborn care. Due to prior participation in a resuscitation clinical trial, both facilities use a battery-operated heart rate meter called NeoBeat (Laerdal Global Health, Stavanger, Norway; Fig 1B) to support the resuscitation of neonates who do not breathe at birth. One of the two facilities offers Cesarean section and has access to physicians as consultants for challenging newborn resuscitations.

## Provider training and provision of resuscitation equipment

Before initiating the registry, all providers participated in a one-day Helping Babies Breathe (HBB) workshop. Local master trainers used adapted HBB 2nd edition training materials in French, which incorporated indications for and use of the Neo-Beat heart rate meter. We instructed providers to place NeoBeat on infants who were not breathing by 30 seconds after delivery. Following initial HBB training, head nurse midwives at each facility encouraged weekly low-dose, high-frequency simulation practice using the NeoNatalie Live mannikin (Laerdal Global Health, Stavanger, Norway) [13,14]. To facilitate the preparation of resuscitation equipment before every birth, we provided each facility with one resuscitation station which was developed and evaluated in the MALA study conducted at Bharatpur Hospital, Nepal (Fig 1C) [15].

## Data collection in the registry

Both participating facilities record care during labor and delivery in a paper delivery registry with standardized fields endorsed by the DRC Ministry of Health. Between September 1st, 2022, and August 31st, 2023, study nurses abstracted demographic, descriptive, and outcome data from this facility delivery registry and individual patient medical records and entered the data into a digital database (KoboCollect, Kobo, Cambridge, Massachusetts, United States). All responses of

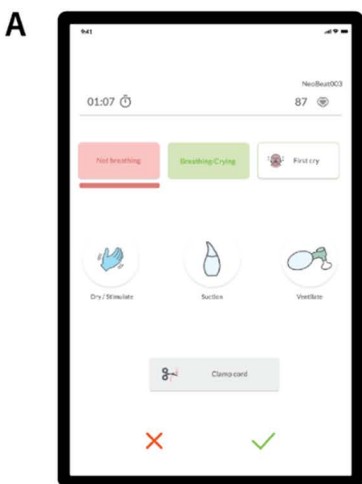
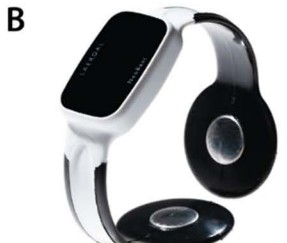
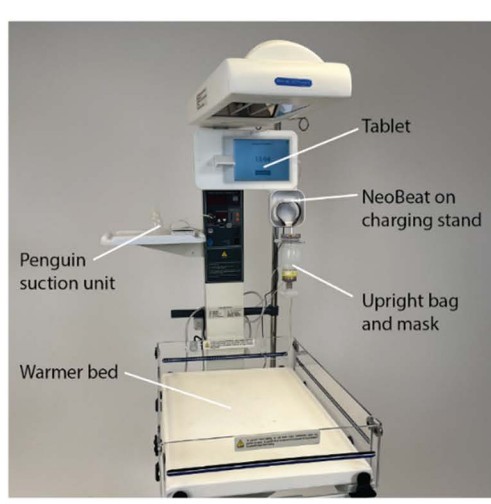

**Fig 1. Registry tools. A)** The **Liveborn Observation application (app)** is a mobile health app used as an observation tool to record the initiation and duration of key resuscitation actions. An observer using the app collects data on the timing of drying/stimulation, suctioning and ventilation for an individual patient at birth. The observer also records the breathing status of the newborn, including the time of the newborn's first cry. **B) NeoBeat** is a reusable battery-operated heart rate meter that can be quickly placed around the newborn's torso by a single provider to gather heart rate data using dry-electrode technology. NeoBeat communicates heart rate to the Liveborn Observation app using Bluetooth technology. **C)** The **Resuscitation station** includes a warmer bed with a penguin suction unit, an upright bag and mask, and a NeoBeat with a charging stand. The embedded tablet can be used to video-record resuscitations.

"unknown" were queried to facilitate optimal data capture. Responses that remained unknown after querying were treated as missing.

We documented the details of resuscitation care by direct observation using the Liveborn Observation app (co-developed by the University of North Carolina at Chapel Hill and Laerdal Medical) [16]. The Liveborn Observation app allows an observer to document newborn resuscitation care at delivery by recording the initiation and duration of key resuscitation actions and the baby's respiratory status (Fig 1A). Study nurses, providers, and others (e.g., environmental health services staff and nurse trainees) used the Liveborn Observation app to register events at birth. The study physician ensured that observers were trained in data collection with the Liveborn Observation app through a guided orientation, practice observing video-recorded resuscitations, and coaching during the observer's use of the app in clinical resuscitations to ensure accurate registration. We supplemented data collected by the Liveborn Observation app with heart rate data from NeoBeat when midwives placed the device.

### Evaluation of registry congruence with ILCOR variables

To evaluate the congruence of our registry with ILCOR variables, we first eliminated all ILCOR variables that were non-applicable to our setting based on report by facility leadership (e.g., therapies not offered or not practiced). For all variables in our registry, we determined the percent completion from the respective data source. For ILCOR-recommended variables not being collected, we annotated the main barrier to collecting (e.g., not reliably recorded in the medical record, not reliably diagnosed). We did not include ILCOR variables for preterm only (≤32 weeks gestational age) in our evaluation of congruence.

### Approvals

The University of North Carolina (UNC) Institutional Review Board (IRB) and the local Kinshasa School of Public Health (KSPH) IRB approved this study (UNC IRB # 22–0851; KSPH IRB # ESP/CE/62b/2023). Providers participated following written informed consent; a complete waiver of consent was granted for newborns.

## Results

In the first year of the registry, September 1st 2022 to August 31st 2023, we enrolled 6,414 newborns (3,414 and 3,000 newborns at the respective facilities; Fig 2). We observed 3,166 neonates at birth with the Liveborn Observation app (49% of enrolled newborns). Observations were conducted by midwives/nurses (71%), study nurses (8%) and others (21%). Of newborns observed with the Liveborn Observation app (n = 3,166), we collected heart rate data on 183 newborns with NeoBeat; this constituted 30% (85/286) of neonates not breathing by 30 seconds after birth and 52% (72/138) of neonates not breathing by 60 seconds after birth.

ILCOR recommends collecting resuscitation data on neonates born with respiratory or cardiac failure who require PPV or chest compressions. In the first year of our open cohort, 60 out of 3,166 observed livebirths (1.9%) received PPV (Fig 2). Additionally, we collected data on 257 neonates not crying by 30 seconds who did not receive PPV, and 13 neonates who were fresh stillbirths. Of the 13 fresh stillborns, four received PPV.

By facility leadership report, we identified 5 (16%) of ILCOR core variables that did not apply to the facilities in our registry (namely, umbilical cord milking [not practiced], CPAP [not offered in the delivery room], respiratory interfaces [limited to facemask only], supplemental oxygen [not offered in the delivery room] and therapeutic hypothermia [therapy not available]). Of the 26 applicable ILCOR core variables, we collected 25% via medical record abstraction, 13% using the Liveborn Observation app, 29% using the delivery registry, and 3% using NeoBeat (Fig 3A). For ILCOR supplemental variables, we identified 10 (24%) that were not applicable (namely, umbilical cord pH [not offered], number of times umbilical cord milked and whether cord was intact [cord milking not practiced], time endotracheal tube placed [intubation not offered], PPV device [only bag and mask available], respiratory settings [only bag and mask available], FiO2 at completion

of resuscitation [oxygen not offered in the delivery room], fluid boluses [not offered in the delivery room], other interventions [not offered in the delivery room], therapeutic hypothermia mode and control mode [therapy not available] and brain injury on neuroimaging [neuroimaging not offered]). Of the 32 applicable ILCOR supplemental variables, we collected 24% with the Liveborn Observation app, 3% with medical record abstraction, and 12% with the facility delivery registry (Fig 3B).

We noted several barriers and facilitators to implementation of this newborn resuscitation registry. Unreliable documentation in the medical record was a barrier to recording meconium-stained fluid. Limited diagnostic tools were a barrier to collecting air leak syndrome (e.g., lack of xrays) and moderate to severe hypoxic ischemic encephalopathy (HIE; e.g., lack of training in neurological examination and no access to blood gas analysis). Self-prescribed medication usage was a barrier to accurately documenting antenatal steroids; while antenatal steroids are reliably recorded in the medical record when given at the facility, patients frequently obtain steroids independently in the community. We also noted an increased burden of time to conduct individual medical record abstraction compared to collecting data from the facility's delivery register. Training our large pool of observers to use the Liveborn Observation app was a barrier; as such, although the Liveborn Observation app facilitates documentation of the start time and duration of cardiac compressions, we chose not to include these variables in our registry due to the rarity of HBB-trained providers using this therapy. Several facilitators were key to implementation of our registry. Strong support from labor and delivery leadership facilitated partnerships with the primary providers who are midwives. Including resuscitation training and the provision of NeoBeat in the registry increased motivation for midwives to participate in data collection.

Table 1 lists all variables in our registry with the source and completeness of the data. The vast majority of our variables not recommended by ILCOR are in the pre-resuscitation and resuscitation domains; we collected these variables using the Liveborn Observation app, including the timing, duration, and frequency of drying/stimulation, suctioning, and PPV. We also collected data on stillbirth, including type, using delivery registry abstraction.

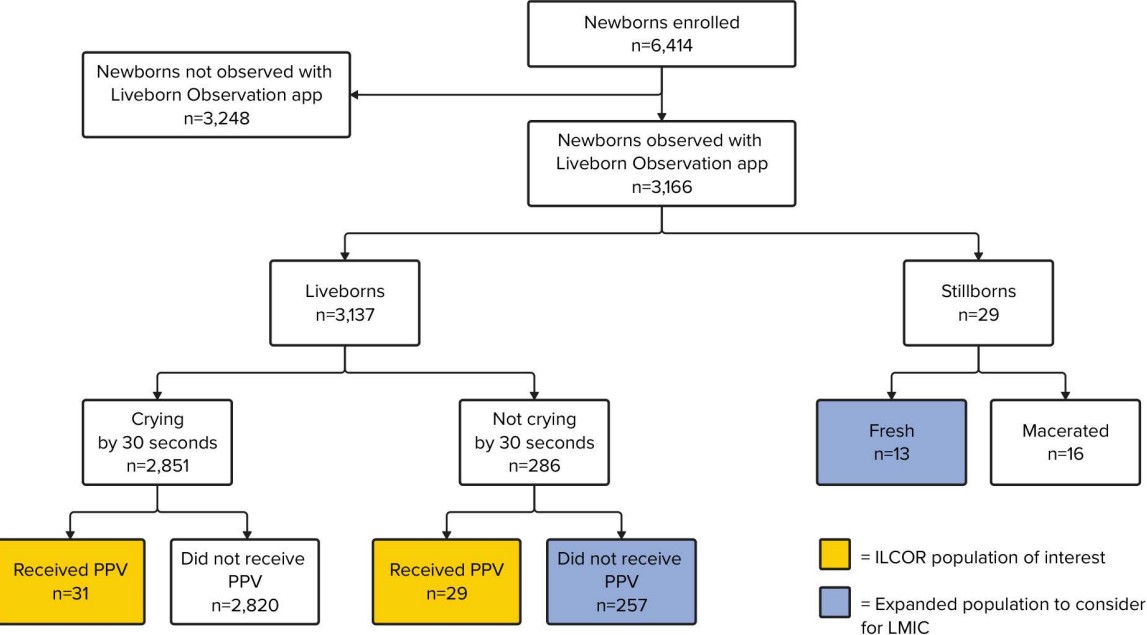

**Fig 2. Registry population.** This figure depicts newborns enrolled in the registry from September 1, 2022, to August 31, 2023. The ILCOR population of interest (depicted in yellow) focuses on liveborns who receive PPV. The expanded population to consider for LMICs (depicted in blue) includes fresh stillborns and those who did not cry by 30 seconds after birth and did not receive PPV. Abbreviations: ILCOR = International Liaison Committee on Resuscitation, PPV = positive pressure ventilation, LMICs = low- and middle-income countries.

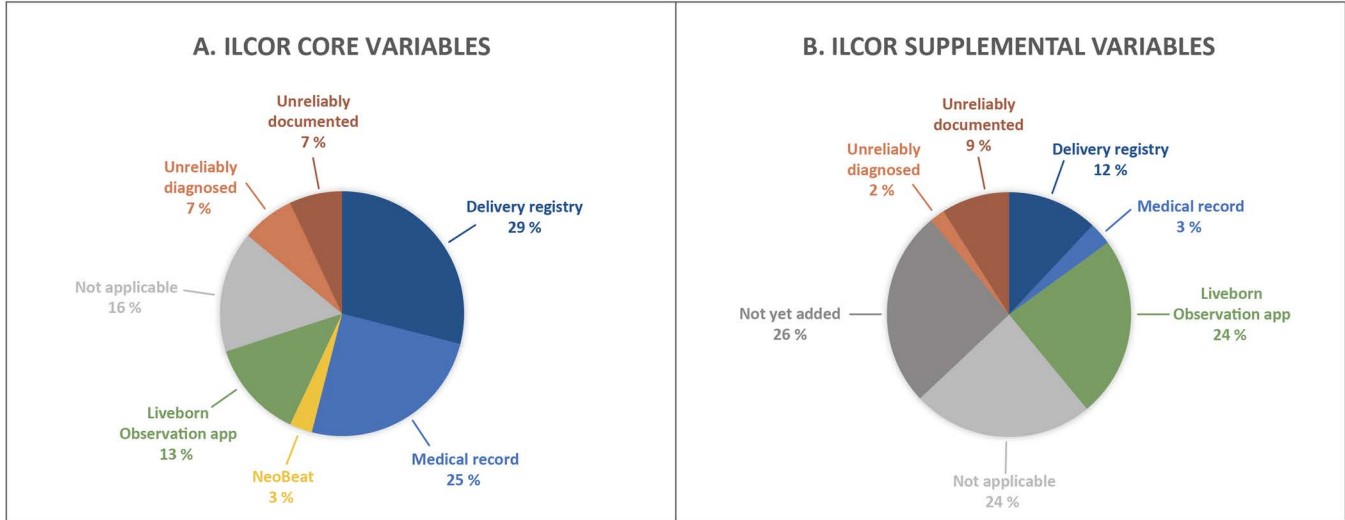

**Fig 3. Data sources for ILCOR-recommended variables in the registry.** After eliminating non-applicable variables per report by facility leadership, we extracted the remaining ILCOR variables using a combination of the facility delivery registry, the medical record, the Liveborn Observation app and NeoBeat (see Table 1). We determined 14% of core variables and 11% of supplemental could not be collected due to either 1) unreliable documentation in the medical record OR 2) unreliable diagnosis as part of standard clinical care. Part A of the figure shows the sources of ILCOR-recommend core variables in our registry and part B shows the sources for ILCOR supplemental variables. Of the 26% of supplemental variables that are not yet added, 18% would be feasible to collect using Liveborn Observation app, 18% would be feasible to collect using the delivery registry and 64% would be feasible to collect using the medical record.

Given our strategy of observing a convenience sample of births, approximately half of participants in the registry have data on resuscitation care. The cohort we observed with the Liveborn app was biased towards a less sick population with lower rates of Cesarean section (0.3% vs 2.2%), prematurity (12.5% vs 17.9% born at 28–36 weeks gestation), low birth weight (9.3% vs 12.8% birth weight 1500-2499g), APGAR 1min ≤ 5 (4.7% vs 7.4%), and neonatal deaths before discharge (0.9% vs 2%) compared to those not observed [S1 Table]. Baseline characteristics by facility are notable for a sicker population at facility 2 with higher rates of Cesarean section (2.6% vs 0%), birth weight ≤ 1499 grams (4.2% vs 0.4%) and 1500–2499 grams (13.6% vs 8.8%), 28–36 weeks gestation (20.1% vs 11%), APGAR 1 min ≤ 5 (8.5% vs 3.9%), and stillbirths (2.9% vs 1.1%) along with a lower rate of neonatal transfer to another facility (0.3% vs 2.1%) [S2 Table].

Table 2 illustrates the data available regarding resuscitation care comparing variables recommended by ILCOR as core, variables as supplemental and additional variables we obtained in our cohort using the Liveborn Observation app. Based only on ILCOR core variables, 1.9% of the cohort received PPV and among neonates who had NeoBeat placed by the midwife, 25% had a heart rate <100 beats per minute at 1 minute after birth. With ILCOR supplemental variables, it can also be noted that 98.9% were stimulated, 73.6% suctioned, and the median time to initiate PPV was 171 seconds (Q1 91, Q3 298) with ventilation lasting a median of 1 minute (Q1 1, Q3 2). By adding variables from the Liveborn Observation app, it can additionally be noted that providers started stimulating at a median 11 seconds after birth (Q1 7, Q3 19) and stimulated for a median of 36 seconds (Q1 20, Q2 56); they started suctioning at a median 39 seconds after birth (Q1 21, Q3 64) and continued for a median 26 seconds (Q1 15, Q3 42); providers spent a median 53 seconds (Q1 35, Q3 132) ventilating, with a median average time between each ventilation episode of 38 seconds (Q1 19, Q3 55); the median heart rate at initiation of PPV was 144 (quartiles 123, 172) and only 8 of 38 cases had a heart rate < 100 bpm when PPV was initiated.

**Table 1. Registry variables in each ILCOR-recommended domain with associated data source and completeness for year one of the registry.**

| | Data Source | Percent of Enrolled Neonates for Whom Variable is Known |
|---|---|---|
| **Setting** | | |
| *ILCOR Core* | | |
| Country | Facility delivery registry | 100 |
| Location (i.e., in-hospital versus out-of-hospital) | Medical record | 100 |
| *ILCOR Supplemental* | | |
| Planned out-of-hospital birth | Medical record | 100 |
| High-risk pregnancy center | National Program of Reproductive Health | 100 |
| **Patient** | | |
| *ILCOR Core* | | |
| Gestational age | Facility delivery registry | 99.5 |
| Multiple gestation | Facility delivery registry | 100 |
| Birth weight | Facility delivery registry | 100 |
| Sex | Facility delivery registry | 100 |
| Major congenital anomaly/genetic syndrome | Facility delivery registry | 100/0 |
| Prenatal care | Facility delivery registry | 0 |
| *ILCOR Supplemental* | | |
| Maternal age | Facility delivery registry | 100 |
| Maternal education | Medical record | 0 |
| Maternal hypertension | Medical record | 0 |
| Maternal diabetes | Medical record | 0 |
| Maternal smoking | Medical record | 0 |
| Type of congenital anomaly/genetic syndrome | Medical record | 100/0 |
| Any prenatal limit to resuscitation | None (not in the medical record) | 0 |
| **Antepartum** | | |
| *ILCOR Core* | | |
| Exposure to antenatal corticosteroids | Medical record | 0 |
| Meconium-stained fluids | Medical record | 0 |
| Cesarean delivery | Facility delivery registry | 100 |
| *ILCOR Supplemental* | | |
| Assisted delivery | Facility delivery registry | 100 |
| Labor before cesarean | Medical record | 0 |
| Emergency cesarean | Medical record | 0 |
| Complete antenatal corticosteroids | Medical record | 0 |
| Antepartum hemorrhage | Facility delivery registry | 0 |
| **Birth/Pre-resuscitation** | | |
| *ILCOR Core* | | |
| Umbilical cord clamping timing | Liveborn Observation app | 47 |
| Heart rate at 1 minute | NeoBeat | 3 |
| Respiratory effort at 1 minute | Liveborn Observation app | 49 |
| Apgar score at 1 minute | Facility delivery registry | 100 |
| Apgar score at 5 minutes | Facility delivery registry | 100 |

*(Continued)*

**Table 1.** (Continued)

|  | Data Source | Percent of Enrolled Neonates for Whom Variable is Known |
|---|---|---|
| *ILCOR Supplemental* |  |  |
| Date of birth | Facility delivery registry | 100 |
| Time of birth | Facility delivery registry | 100 |
| Time between birth and cord clamping | Liveborn Observation app | 47 |
| Initial steps: stimulation | Liveborn Observation app | 49 |
| Initial steps: suction | Liveborn Observation app | 49 |
| Initial steps: thermoregulation | Liveborn Observation app | 49 |
| *Additional Variables in the Registry* |  |  |
| Time to first cry | Liveborn Observation app | 49 |
| Number of suctioning episodes | Liveborn Observation app | 49 |
| Initiation time of each suctioning episode | Liveborn Observation app | 49 |
| Duration of each suctioning episode | Liveborn Observation app | 49 |
| Number of drying/stimulation episodes | Liveborn Observation app | 49 |
| Initiation time of each drying/stimulation episode | Liveborn Observation app | 49 |
| Duration of each drying/stimulation episode | Liveborn Observation app | 49 |
| **Resuscitation** |  |  |
| *ILCOR Core* |  |  |
| PPV | Facility delivery registry | 100 |
| Cardiac compressions | Medical record | 0 |
| Epinephrine | Medical record | 0 |
| *ILCOR Supplemental* |  |  |
| Number of providers caring for newborn | Liveborn Observation app | 0 |
| Time PPV started | Liveborn Observation app | 49 |
| Duration of PPV (i.e., first commenced to last performed or resuscitation ended) | Liveborn Observation app | 49 |
| Time chest compressions started | Liveborn Observation app | 0 |
| Duration of chest compressions | Liveborn Observation app | 0 |
| Support at completion of resuscitation | Liveborn Observation app | 49 |
| Epinephrine: number of doses, route, dose, timing | None (not in medical record) | 0 |
| *Additional Variables in the Registry* |  |  |
| Number of PPV episodes | Liveborn Observation app | 49 |
| Initiation time of each PPV episode | Liveborn Observation app | 49 |
| Duration of each PPV episode | Liveborn Observation app | 49 |
| **Post-resuscitation** |  |  |
| *ILCOR Core* |  |  |
| Immediate disposition | Facility delivery registry | 0 |
| Temperature at 1 hour | Medical record | 0 |
| *ILCOR Supplemental* |  |  |
| Transfer to higher level of care | Facility delivery registry | 99.9 |
| Glucose: measured and lowest value | Medical record | 0 |
| **Outcomes** |  |  |
| *ILCOR Core* |  |  |

*(Continued)*

**Table 1.** (Continued)

|  | Data Source | Percent of Enrolled Neonates for Whom Variable is Known |
|---|---|---|
| Death in initial resuscitation area | Medical record AND/OR Live-born Observation app | 100 |
| Death before hospital discharge | Medical record | 99.9 |
| Duration of hospital stay | Medical record | 99.8 |
| Air leak | Medical record | 0 |
| Moderate to severe encephalopathy | Medical record | 0 |
| *ILCOR Supplemental* |  |  |
| Death after discharge before the last follow-up | None | 0 |
| Meconium aspiration syndrome | Medical record | 0 |
| *Additional Variables in the Registry* |  |  |
| Stillbirth | Facility delivery registry AND/OR Liveborn Observation app | 100 |
| Fresh vs macerated stillbirth | Facility delivery registry AND/OR Liveborn Observation app | 100 |
| Neonatal death in the first 24 hours after birth | Facility delivery registry AND/OR Liveborn Observation app | 0 |

## Discussion

We developed a newborn resuscitation registry in two birth facilities in Kinshasa, DRC using a combination of delivery registry and medical record abstraction, direct observation using the Liveborn Observation app, and heart rate using NeoBeat. Our dataset incorporates 85% of applicable ILCOR core variables. The registry includes additional variables beyond those recommended by ILCOR that allow for thorough identification of gaps in quality resuscitation pertinent to low-resource settings.

Our experience establishing a registry with novel tools provides a model for implementing ILCOR's guidelines for uniform reporting of clinical resuscitation studies in low-resource settings. We found that a combination of abstraction from facility records and direct observation was needed to obtain critical data relevant to neonatal resuscitation. The facility delivery registry endorsed by the DRC Ministry of Health for data collection in all birth facilities included 29% of ILCOR core variables and 12% of ILCOR supplemental variables. This suggests that collecting ILCOR-recommended variables in many low-resource settings will require abstraction from medical records which have varying standards for the amount and quality of data. Paper-recording keeping in LMICs adds to the burden of data collection for a registry; investment in a standardized electronic delivery register and/or medical record could improve completeness of the data and reduce the burden of data collection. While limited quality and content of medical record data may be common barriers to implementing ILCOR's guidelines in low-resource settings [18], we found high rates of facility documentation in the medical record for several ILCOR core variables not initially in our registry. This made it feasible to expand the registry to include three additional core variables from the medical record (cardiac compressions, epinephrine, temperature at 1 hour) with feasibility to further expand by adding eight supplemental variables from the medical record. For supplemental ILCOR variables, the Liveborn Observation app was particularly key; of the 13 applicable supplemental variables in the birth/pre-resuscitation and resuscitation domains, we collected 10 (77%) using the app. While we did not collect data on chest compressions, these could be documented using additional features embedded in the Liveborn Observation app.

Based on common gaps in quality resuscitation care in low-resource settings, we identified key additional variables in the birth/pre-resuscitation and resuscitation domains beyond those recommended by ILCOR that are important indicators

**Table 2. Pertinent indicators beyond those recommended by ILCOR in the birth/pre-resuscitation and resuscitation domains.**

| Action | Category | Pertinent Indicators | Newborns, n = 3,166 Median (Quartiles) |
|---|---|---|---|
| **Birth/pre-resuscitation** | | | |
| Stimulation | ILCOR Core | n/a | |
| | ILCOR Supplemental | Stimulated, n (%) | 3130 (98.9) |
| | Liveborn Observation app | Number of episodes per newborn | 1 (1, 1) |
| | | Initiation time from birth [17] | 11 (7, 19) |
| | | Total duration of all episodes [s] | 36 (20, 56) |
| Suctioning | ILCOR Core | n/a | |
| | ICOR Supplemental | Suctioned, n (%) | 2330 (73.6) |
| | Liveborn Observation app | Number of episodes per newborn | 1 (1, 1) |
| | | Initiation time from birth [s] | 39 (21, 64) |
| | | Total duration of all episodes [s] | 26 (15, 42) |
| Heart Rate | ILCORE Core: Heart rate at 1 minute | Heart rate at 1 minute ≥100 bpm, n (%) | 86 (2.7) |
| | | Heart rate at 1 minute <100 bpm, n (%) | 29 (0.9) |
| | | Heart rate at 1 minute undetectable or not measured, n (%) | 3051 (96.4) |
| | NeoBeat | Time to heart rate ≥100 bpm among those with first heart rate <100 bpm [s] (n = 49) | 25 (12, 72) |
| | | First heart rate detected (bpm) (n = 176) | 129 (86, 159) |
| | | Time to first heart rate detected (s) (n = 176) | 36 (5, 68) |
| | | Last heart rate detected (bpm) (n = 176) | 159 (137, 180) |
| | | Time to last heart rate detected (s) (n = 176) | 189 (123, 336) |
| | | Time with HR measurement (s) (n = 176) | 133 (81, 238) |
| | | Heart rate at 30 s (bpm) (n = 70) | 138 (96, 164) |
| | | Heart rate at 60 s (bpm) (n = 115) | 141 (100, 168) |
| | | Heart rate at initiation of PPV (bpm) (n = 38) | 144 (124, 173) |
| **Resuscitation** | | | |
| Positive-pressure ventilation | ILCOR Core | Ventilated, n (%) | 64 (2.0) |
| | ILCOR Supplemental | Initiation time from birth [s] | 171 (91, 298) |
| | | Total duration of all episodes [min] | 1 (1, 2) |
| | Liveborn Observation app | Number of episodes per newborn | 1 (1, 2) |
| | | Duration of first episode [s] | 47 (22, 92) |
| | | Total duration of all episodes [s] | 53 (35, 131) |
| | | Average duration of each episode [s] | 48 (22, 93) |
| | | Average time between each episode [s] | 38 (19, 55) |

*Abbreviations*: s=seconds, bpm=beats per minute, min=minutes.

to collect [19–21]. In particular, delayed initiation of PPV and interruption of PPV are common in low-resource settings with significant impact on neonatal outcomes. For every 30-second delay in PPV, the risk of death or prolonged hospitalization increases 16% [22]. We observed that delayed PPV and pauses in PPV were accompanied by extended duration of stimulation and suctioning, supporting our previous observations in the DRC [1]. Similar observations have been

**Table 3. Considerations for use of ILCOR-recommended guidelines for uniform reporting of neonatal resuscitation in LMICs.**

| Neonatal Utstein Element | Current ILCOR Guideline | Additions to Consider for Research in LMICs |
|---|---|---|
| Eligible resuscitation events | Respiratory and cardiac failure leading to a clinical decision to provide PPV or chest compressions | Newborns (liveborn or fresh stillborn) not crying by 30 s after birth who do not receive PPV or chest compressions |
| Setting | *ILCOR Supplemental* | |
| | Out-of-hospital: planned out-of-hospital birth (Y/N)<br>In-hospital: high-risk pregnancy center (Y/N) | Type (e.g., physician, midwife, nurse) and training level (e.g., doctorate, midwifery, nursing) of providers responsible for resuscitation<br>Type of fetal heart rate monitoring (e.g., continuous vs intermittent; if intermittent, electronic vs fetoscope) |
| Antepartum | *ILCOR Supplemental* | |
| | Antepartum hemorrhage (Y/N) | Acute perinatal event (e.g., cord prolapse, uterine rupture, shoulder dystocia, severe fetal heart rate abnormality) (Y/N) |
| Birth and pre-resuscitation | *ILCOR Core* | |
| | Respiratory effort at 1 min (adequate/inadequate/apneic) | Time to first cry (s) |
| | *ILCOR Supplemental* | |
| | Stimulated (Y/N)<br>Suctioned (Y/N) | Duration of stimulation prior to start of PPV (s)<br>Duration of suctioning prior to start of PPV (s)<br>Duration of stimulation between start and end of PPV (s)<br>Duration of suctioning between start and end of PPV (s) |
| Resuscitation | *ILCOR Supplemental* | |
| | Time PPV started (min:s)<br>Duration of PPV (min) | Duration of each PPV episode (s)<br>Time between each PPV episode (s) |
| Outcomes | *ILCOR Core* | |
| | Death in initial resuscitation area (Y/N)<br>Death before hospital discharge (Y/N) | Stillbirth (Y/N)<br>Neonatal death in the first 24 hours after birth (Y/N) |
| | *ILCOR Supplemental* | |
| | Death before last follow-up (Y/N) | Stillbirth type (e.g., fresh, macerated) |

Abbreviations: PPV = positive pressure ventilation, s = seconds, Y/N = Yes/No, min:s = minutes:seconds, min = minutes.

reported in Tanzania [23,24], and Uganda [25]. These findings have implications for recommendations globally, calling for de-emphasis of suctioning and more emphasis on continued PPV. ILCOR's current recommendation of collecting simply whether or not stimulation and suctioning were performed does not permit the discovery of potential factors contributing to delayed and interrupted PPV in low-resource settings. Furthermore, although pauses in ventilation are common, ILCOR's recommended variables for PPV [time PPV started (min:sec) and the total duration of PPV (expressed in whole minutes)] do not adequately measure this gap. At a minimum, we recommend considering the addition of average duration of each PPV episode (in seconds) and average time between each PPV episode (in seconds) as supplemental variables (Table 3). To understand why ventilation is delayed or interrupted, we also recommend considering the following variables: total duration of stimulation prior to initiation of PPV, total duration of suctioning prior to initiation of PPV, total duration of stimulation between initiation and end of PPV, and total duration of suctioning between initiation and end of PPV (Table 3). To acquire this level of detail, data collection during direct observation would be necessary, and this type of data collection is burdensome. The impact of this practical constraint can be minimized by observing a subset of deliveries. Finally, the type of quality gaps common to a particular setting may be linked to the qualifications of the provider (e.g., skilled birth attendant, nurse, midwife, general practitioner, pediatrician, neonatologist); a facility-level variable

capturing the type of provider responsible for resuscitating newborns could be considered as an addition to the ILCOR supplemental variables in the setting domain (Table 3).

We captured respiratory effort at one minute (core variable in the ILCOR birth/pre-resuscitation domain) using the Liveborn Observation app. While providers are assessing respiratory status with their assignment of a one-minute Apgar score, we found the components of the Apgar score are not recorded in the medical record. Furthermore, we recognize evaluation of breathing status can be a complex and subjective skill, particularly when differentiating between adequate and inadequate breathing. An alternative, and more objective measure of breathing is crying. An observational study in Nepal showed that non-crying after birth had a 100% sensitivity for non-breathing, and that non-crying but breathing infants had almost 12-fold odds of pre-discharge mortality [26]. We suggest that the time to first cry may be a more objective variable for core measurements rather than respiratory effort at one minute (Table 3). Newly-recommended quality process indicators for evaluation of newborn resuscitation in low-resource settings focus on crying rather than breathing due to the subjectivity and complexity of respiratory evaluation [27].

Within the outcomes domain, our registry includes two key variables not included in ILCOR's recommended variables: stillbirth (fresh versus macerated) and neonatal death in the first 24 hours. In low-resource settings, flaccid liveborn infants may be misclassified as stillborn [21,28–30]. Infrequent early and inaccurate heart rate detection after the birth of a non-breathing neonate may contribute to misclassification. Misclassification of a newborn as stillborn rather than liveborn may bias providers towards limited or no resuscitation. [29] To adequately evaluate resuscitation care of depressed, liveborn neonates who may be inaccurately classified as fresh stillbirth in low-resource settings, we strongly recommend the addition of stillbirth, including type, to the outcome domain (Table 3). Early perinatal mortality, defined as neonatal mortality in the first 24 hours after birth and fresh stillbirths, should be considered as an additional variable that averts the problem of misclassification of stillbirth (Table 3).

ILCOR's reporting guideline focused on liveborn neonates who received resuscitation, defined as those with respiratory failure requiring PPV, bradycardia, and cardiac arrest. We strongly recommend expanding what is considered 'resuscitation,' particularly in low-resource settings where PPV for non-breathing newborns at birth may be underutilized. In Nepal, a prospective observational study showed a significant increase in the initiation of PPV for non-crying infants following resuscitation quality improvement interventions (OR 1.22 [95% CI 1.04–1.44]) [31]. In the DRC, we found that ventilation was provided to only 20% of newborns who were not breathing well by 60 seconds after birth. (1) This suggests that we cannot assume that needing PPV is equivalent to receiving PPV in low-resource settings. Due to the concern for underuse of PPV in low-resource settings, we recommend considering expanding the target population to include all newborns not crying by 30 seconds after birth (Table 3). Ministries of Health may similarly consider expanding the target population for evaluating quality resuscitation given underuse of PPV in low-resource settings.

Reducing morbidity and mortality from intrapartum-related events requires moving beyond a focus on postnatal resuscitation to include an understanding of fetal well-being. In reflecting on the ILCOR antepartum domain, we noted key details regarding fetal well-being and acute perinatal events are not included. While ILCOR recommends collecting data on maternal hemorrhage, other key events in the intrapartum period, such as cord prolapse, uterine rupture, obstructed labor, and severe fetal heart rate abnormalities, are not included. Such intrapartum events significantly increase the risk of respiratory depression and neonatal encephalopathy; for example, in Uganda, hemorrhage, cord prolapse, or uterine rupture significantly increased the risk of neonatal encephalopathy (adjusted OR 8.74 [95% CI 1.70–45.02]) [32]. While we aspire to add more granular data on the intrapartum period to our registry, we recognize the high burden of collecting this data in settings with limited diagnostic tools, including limited use of continuous fetal heart rate monitoring. Given current norms regarding intrapartum monitoring in low-resource settings, we do not recommend these variables for inclusion in ILCOR's core variables but suggest they could be considered as supplemental. Type of fetal heart rate monitoring (e.g., continuous or intermittent? electronic or fetoscope?) could be added to the ILCOR supplemental variables in the setting domain to

clarify the level of intrapartum care provided (Table 3). Intrapartum data is critical for identifying and evaluating strategies to prevent HIE; creative and low-cost strategies for gathering this data in low-resource settings are needed.

Our strategy for implementing a resuscitation registry in the DRC has several limitations. While direct observation was a key part of our strategy for data collection, it is burdensome and requires continual staff presence due to the unpredictability of when births will occur. We lowered the burden by observing only a convenience sample; however, drawing inferences from a convenience sample introduces the potential for selection bias. The convenience sample we observed was biased towards a less sick population, likely due to less availability for midwives to observe during more acute deliveries. The study design did not include comparative analysis, which limits our understanding of the effect of establishing the registry. The use of NeoBeat was low (only 3.6% of infants at one minute after birth had a measured heart rate), which makes inferences from heart rate data challenging in this cohort. Midwives building the habit of using NeoBeat has been challenging, and we are exploring midwives' preferences of using it on every baby or only ill infants. The burden of cleaning and charging the device is also an important consideration. Given the very early time to first heart rate detected for the first quartile (median 35 seconds [Q1 5, Q3 68]) we suspect that some registrations with the Liveborn Observation app were initiated some seconds after birth. While the data collected with the Liveborn Observation app are likely more precise than recall, because observers may also have clinical responsibilities, competing tasks may influence the accuracy of the data. We did not perform quality checks or evaluate inter-rater reliability among observers. Since the initial year of this registry, we have restricted data collection with the Liveborn Observation app to study nurses only. Exclusive research staff for observations, however, significantly increases the resources involved. It is important to note that our experience implementing a resuscitation registry in the DRC may not be generalizable to all low-resource settings.

Since the fall of 2023, we have incorporated audio-video recording for resuscitations occurring at the warmer using a digital camera secured within a case to standardize the area of footage captured. The observer initiates audio-video recording via the Liveborn Observation app. We anticipate that audio-video recording will have several potential advantages. Future directions for the registry include validation of the accuracy of observational data compared to audio-video recording, and inclusion of audio-video recording data on corrective actions during PPV, competing tasks, and team communication.

## Conclusions

We established an open cohort study focused on newborn resuscitation in two delivery facilities in Kinshasa, DRC, using a combination of delivery registry, medical record abstraction, observation of deliveries, and heart rate monitoring. Our dataset includes 85% of applicable ILCOR-recommended core variables and additional contextually important variables to evaluate resuscitation care in low-resource settings. Our experience establishing a resuscitation registry with novel tools in the DRC serves as a model for resuscitation research in low-resource settings and provides important insight to support subsequent versions of ILCOR's guidelines.

## Supporting information

**S1 Accessible Data. LEARN Registry Manuscript Data for Sharing.**
(XLSX)

**S1 Inclusivity in Global Research. Inclusivity in Global Research.**
(DOCX)

**S1 Table. Maternal and neonatal characteristics observed and not observed with the Liveborn Observation.**
(DOCX)

**S2. Table. Maternal and neonatal characteristics cumulative and divided by facility.**
(DOCX)

## Acknowledgments

We want to thank the midwives, study nurses, environmental health services staff, and trainees at the two health facilities in Kinshasa, DRC, who observed births with the Liveborn Observation App.

## Author contributions

**Conceptualization:** Melissa Bauserman, Carl Bose, Jackie K. Patterson.

**Data curation:** Daniel Ishoso, Eric Mafuta, Joar Eilevstjønn.

**Formal analysis:** Amy Mackay, Joar Eilevstjønn, Jackie K. Patterson.

**Funding acquisition:** Jackie K. Patterson.

**Investigation:** Amy Mackay, Daniel Ishoso, Patricia Gomez, Waldemar Carlo, Melissa Bauserman, Jackie K. Patterson.

**Methodology:** Amy Mackay, Daniel Ishoso, Eric Mafuta, Joar Eilevstjønn, Patricia Gomez, Melissa Bauserman, Carl Bose, Jackie K. Patterson.

**Project administration:** Daniel Ishoso, Eric Mafuta, Jackie K. Patterson.

**Resources:** Daniel Ishoso, Eric Mafuta, Carl Bose, Jackie K. Patterson.

**Software:** Daniel Ishoso, Eric Mafuta, Joar Eilevstjønn, Jackie K. Patterson.

**Supervision:** Carl Bose, Jackie K. Patterson.

**Validation:** Jackie K. Patterson.

**Writing – original draft:** Amy Mackay, Jackie K. Patterson.

**Writing – review & editing:** Amy Mackay, Daniel Ishoso, Eric Mafuta, Joar Eilevstjønn, Patricia Gomez, Waldemar Carlo, Melissa Bauserman, Carl Bose, Jackie K. Patterson.

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
