## [Decision Letter · Decision Letter 0]

21 Sep 2024

PONE-D-24-30580Establishment of a neonatal resuscitation registry in the Democratic Republic of the Congo: An open cohort studyPLOS ONE

Dear Dr. Patterson,

Thank you for submitting your manuscript to PLOS ONE. After careful consideration, we feel that it has merit but does not fully meet PLOS ONE’s publication criteria as it currently stands. Therefore, we invite you to submit a revised version of the manuscript that addresses the points raised during the review process.

As also suggested by the reviewers, minor revisions to the manuscript are necessary. I would especially also find it interesting to know why only one co-author of the manuscript seems to be really affiliated with the region you are writing about?

We look forward to receiving your revised manuscript.

Kind regards,

Sebastian Schnaubelt, MD, PhD

Academic Editor

PLOS ONE

Journal Requirements:

Reviewers' comments:

Reviewer's Responses to Questions

**Comments to the Author**

1. Is the manuscript technically sound, and do the data support the conclusions?

Reviewer #1: Yes

2. Has the statistical analysis been performed appropriately and rigorously? 

Reviewer #1: Yes

3. Have the authors made all data underlying the findings in their manuscript fully available?

Reviewer #1: Yes

4. Is the manuscript presented in an intelligible fashion and written in standard English?

Reviewer #1: Yes

5. Review Comments to the Author

Reviewer #1: The current study is on a topic of relevance and general interest to the readers of the journal. I found the paper to be overall well written and felt confident that the authors performed careful the preparation of the manuscript. However, I still have some comments and thoughts, I want to share with the authors to further improve their paper:

- Following references are missing: First two sentences of the introduction

- Row 52: ... critical improving / to improve .. not to improving

- Fig. 2: Why didn't you include newborns not crying by 60 sec in the group of interest for ILCOR?

- I would put all footnotes next to the variable name and not next to the number to get consistency

- Row 194: you are not able to infer from Fig 3b on the origin of data

- Tab 2:

 For the variable "Heart Rate -> NeoBeat -> Heart rate at initiation of PPV" did you only use data from the NeoBeat device or also from the Liveborn Observation app?

 Is there a difference between the variables "PPV -> ILCOR Supplemental -> Total duration of all episodes [min]" and "PPV -> Liveborn Observation app -> Total duration of all episodes [s]" beside from the unit?

- Tab 3: are you suggesting the adaptations as additional variables or to adapt the current recommendations

- Please add the following points: limitations, knowledge gaps and next steps

- Please try to include the following literature:

DOI: 10.1016/S2214-109X(23)00302-9

DOI: 10.1542/peds.2020-016915G

- One point of criticism is that only one of the authors is affiliated with a clinic in DRC. To create a sustainable research environment in LMICs, local researches should be encouraged and supported to do their own research projects. By including local researches in developing such projects, know-how in LMICs can be build up and even more researcher have the opportunity to find solutions in this field of interest

6. PLOS authors have the option to publish the peer review history of their article (what does this mean? ). If published, this will include your full peer review and any attached files.

**Do you want your identity to be public for this peer review?** For information about this choice, including consent withdrawal, please see our Privacy Policy .

Reviewer #1: No

---

## [Author Response · Author response to Decision Letter 0]

12 Nov 2024

October 28, 2024

Dr. Ribka Amsalu

Academic Editor

PLOS ONE

Dear Dr. Amsalu,

Thank you for your further consideration of our manuscript titled “Establishment of a neonatal resuscitation registry in the Democratic Republic of the Congo: An open cohort study” (PONE-D-24-30580). We thank you and the reviewer for the helpful comments, and respectfully submit the final revised manuscript. Below is a point-by-point response to the reviewer’s comments.

Sincerely,

Jackie K. Patterson, MD, MPH

Associate Professor of Pediatrics

Division of Neonatal-Perinatal Medicine

University of North Carolina at Chapel Hill

Response to Reviewer

The current study is on a topic of relevance and general interest to the readers of the journal. I found the paper to be overall well-written and felt confident that the authors performed careful the preparation of the manuscript. However, I still have some comments and thoughts, I want to share with the authors to further improve their paper:

1) Following references are missing: First two sentences of the introduction

Response:

We added Oza et al. to the first sentence of the introduction, Bellad et al., Msemo et al., and Versantvoort et al. to the second sentence of the introduction.

2) Row 52: ... critical improving / to improve .. not to improving

Response:

We revised per your suggestion.

3) Fig. 2: Why didn't you include newborns not crying by 60 sec in the group of interest for ILCOR?

Response:

The ILCOR population of interest is focused on the provision of PPV and not on those with apnea or insufficient respirations who do not receive PPV. We recognize that Figure 2 may have been confusing as we displayed those not crying by 30 seconds and those not crying by 60 seconds which are not mutually exclusive groups. We have deleted the box referring to those not crying by 60 seconds for clarity. As resuscitation algorithms emphasize initiating ventilation by 60 seconds after birth, we recommended including all newborns not crying by 30 sec.

4) I would put all footnotes next to the variable name and not next to the number to get consistency

Response:

We revised the footnotes in Table 1 to be next to the variable for consistency.

5) Row 194: you are not able to infer from Fig 3b on the origin of data

Response:

We added a reference to Table 1 in the Figure 3 legend to add transparency about the data sources.

6) Table 2: For the variable "Heart Rate -> NeoBeat -> Heart rate at initiation of PPV" did you only use data from the NeoBeat device or also from the Liveborn Observation app?

Response:

For Table 2, all heart rate data obtained in this cohort comes from NeoBeat. We only collected heart rate data for cases where the Liveborn Observation app was used. NeoBeat connects via Bluetooth to the Liveborn Observation app and the data is then downloaded from the Liveborn Observation app.

7) Table 2: Is there a difference between the variables "PPV -> ILCOR Supplemental -> Total duration of all episodes [min]" and "PPV -> Liveborn Observation app -> Total duration of all episodes [s]" beside from the unit?

Response:

Yes, the difference between “Total duration of all episodes [min]” in ILCOR supplemental and “Total duration of all episodes [s]” is the unit. Recording duration in whole units of minutes does not give enough granular data to understand gaps in care related to interruptions in PPV. Additions in the text were made in lines 268-271 to help clarify this distinction for readers.

8) Table 3: are you suggesting the adaptations as additional variables or to adapt the current recommendations

Response:

In Table 3, we suggest additional variables to consider. Table 3 column header and first recommended addition were revised for clarity.

9) Please add the following points: limitations, knowledge gaps and next steps

Response:

We discuss the limitations of our study in the second to last paragraph of the discussion; we have clarified the content of this paragraph with a new topic sentence. We discuss knowledge gaps and next steps in the last paragraph of the discussion, and have added the following content:

“Future directions for the registry include validation of the accuracy of observational data compared to audio-video recording, and inclusion of audio-video recording data on corrective actions during PPV, competing tasks, and team communication.” (Lines 351-353)

10) Please try to include the following literature:

DOI: 10.1016/S2214-109X(23)00302-9 Schnaubelt et al in 3rd paragraph of introduction

DOI: 10.1542/peds.2020-016915G Keenan et al in 2nd paragraph of introduction

Response:

We incorporated references Keenan et al. and Schnaubelt et al. in the introduction (Line 72).

11) One point of criticism is that only one of the authors is affiliated with a clinic in DRC. To create a sustainable research environment in LMICs, local researches should be encouraged and supported to do their own research projects. By including local researches in developing such projects, know-how in LMICs can be build up and even more researcher have the opportunity to find solutions in this field of interest

Response:

The University of North Carolina at Chapel Hill and the Kinshasa School of Public Health have had a twenty-year partnership conducting research for maternal, newborn and child health in the DRC, including through the NICHD Global Network. The registry was developed and implemented by investigators from both institutions. Drs. Ishoso and Mafuta are key members of this team and both are listed as co-authors. We noted that PLOS One allows joint first authorship; as such, we have listed Daniel Ishoso as joint first author with Amy Mackay.

---

## [Decision Letter · Decision Letter 1]

11 Dec 2024

PONE-D-24-30580R1Establishment of a neonatal resuscitation registry in the Democratic Republic of the Congo: An open cohort studyPLOS ONE

Dear Dr. Patterson,

Thank you for submitting your manuscript to PLOS ONE. After careful consideration, we feel that it has merit but does not fully meet PLOS ONE’s publication criteria as it currently stands. Therefore, we invite you to submit a revised version of the manuscript that addresses the points raised during the review process.

We look forward to receiving your revised manuscript.

Kind regards,

Sebastian Schnaubelt, MD, PhD

Academic Editor

PLOS ONE

Journal Requirements:

Reviewers' comments:

Reviewer's Responses to Questions

**Comments to the Author**

1. If the authors have adequately addressed your comments raised in a previous round of review and you feel that this manuscript is now acceptable for publication, you may indicate that here to bypass the “Comments to the Author” section, enter your conflict of interest statement in the “Confidential to Editor” section, and submit your "Accept" recommendation.

Reviewer #2: (No Response)

Reviewer #3: (No Response)

2. Is the manuscript technically sound, and do the data support the conclusions?

Reviewer #2: Yes

Reviewer #3: Partly

3. Has the statistical analysis been performed appropriately and rigorously? 

Reviewer #2: Yes

Reviewer #3: N/A

4. Have the authors made all data underlying the findings in their manuscript fully available?

Reviewer #2: Yes

Reviewer #3: Yes

5. Is the manuscript presented in an intelligible fashion and written in standard English?

Reviewer #2: Yes

Reviewer #3: Yes

6. Review Comments to the Author

Reviewer #2: 1. Significance and Novelty:

The paper presents a significant contribution to neonatal care research, particularly in low-resource settings like the Democratic Republic of the Congo (DRC). The establishment of a neonatal resuscitation registry using novel tools such as the Liveborn Observation application and NeoBeat is innovative and addresses a critical gap in understanding real-time resuscitation practices. The registry's alignment with the International Liaison Committee on Resuscitation (ILCOR) recommendations enhances its relevance and potential for global impact. The detailed account of the registry's development, including provider training and the adaptation of tools for the local context (e.g., sections 104-113), demonstrates a thoughtful approach to addressing the challenges of neonatal care in low-resource settings.

2. Strengths:

The methodology employed in this study, including the use of digital tools for direct observation and data collection, represents a robust approach to capturing detailed information on neonatal resuscitation practices. This methodological rigor is a key strength of the paper.

The relevance of this research to global health, especially in improving neonatal outcomes in low-resource settings, cannot be overstated. The paper's focus on a real-world application of ILCOR guidelines and the identification of gaps in current practices is particularly commendable (e.g., sections 77-78).

The comprehensive data collection spanning over a year and covering a significant number of newborns (6,414 newborns with observational data on 3,159) provides a solid foundation for the study's conclusions and recommendations (e.g., section 33-34).

3. Weaknesses:

The paper lacks a comparative analysis or control group, which may limit the ability to attribute observed outcomes directly to the interventions or registry implementation.

While the paper mentions the congruence of the registry with ILCOR variables, there is limited discussion on the implications of these findings for neonatal care practices or policy recommendations beyond the study context.

Some sections of the paper are heavily descriptive, particularly in the introduction and methodology sections, which might detract from the focus on the study's outcomes and implications.

The discussion on barriers and facilitators to resuscitation data collection is somewhat brief and lacks depth. Given the novel approach of this registry, a more detailed exploration of these aspects would be valuable for replication in other settings.

4. Suggestions for Improvement:

Introduce a comparative analysis where possible, or discuss the limitations of the study design regarding the absence of a control group. This could help contextualize the findings within the broader literature on neonatal resuscitation practices.

Expand the discussion on the implications of registry findings for neonatal care practices and policy, both within the DRC and globally. Highlighting specific variables that were not collected but are critical for improving neonatal outcomes could be particularly insightful.

Streamline the introductory and methodology sections to maintain a sharper focus on the study's objectives, findings, and implications. Reducing descriptive content that does not directly contribute to understanding the study's impact could enhance readability and engagement.

Provide a more detailed analysis of the barriers and facilitators encountered during the registry's implementation, including how these challenges were addressed or could be overcome in future research. Sharing lessons learned in this aspect could be extremely valuable to others looking to implement similar registries in low-resource settings.

Summary Statement:

This paper represents an important step forward in understanding and improving neonatal resuscitation practices in low-resource settings. The novel use of digital tools for data collection and the registry's alignment with international recommendations are notable strengths. However, to fully realize its potential impact, the paper could benefit from a more focused discussion on the implications of its findings, as well as a deeper analysis of the challenges encountered during implementation. With these enhancements, the paper could make a significant contribution to global health research and practice.

Reviewer #3: Thank you for your submission.

It is crucial that feedback is provided to the ILCOR neonatal group so that any revision of the Utstein criteria can be updated whenever appropriate.

It is however “the hope of the Neonatal Utstein Working Group that, at minimum, all specified core data elements will be reported using the standardized definitions provided here.”

It would be useful to provide feedback not only about what data was difficult to collect, but in what way (if any) the authors planned to explore collecting more of the core data elements. Barriers to collection are crucial to discuss and solutions may need to be creative.

If this is not possible, it may be that the key components of this manuscript would be better published as a letter.

7. PLOS authors have the option to publish the peer review history of their article (what does this mean? ). If published, this will include your full peer review and any attached files.

**Do you want your identity to be public for this peer review?** For information about this choice, including consent withdrawal, please see our Privacy Policy .

Reviewer #2: **Yes: ** Olivier Mukuku

Reviewer #3: **Yes: ** Peter Morley

---

## [Author Response · Author response to Decision Letter 1]

24 Jan 2025

January 24, 2025

Sebastian Schnaubelt, MD, PhD

Academic Editor

PLOS ONE

Dear Dr. Schnaubelt,

Thank you for your continued consideration of our manuscript, “Establishment of a neonatal resuscitation registry in the Democratic Republic of the Congo: An open cohort study” (PONE-D-24-30580). Following is a point-by-point response to the reviewer’s comments to match the revised manuscript we have submitted.

Sincerely,

Jackie K. Patterson, MD, MPH

Associate Professor of Pediatrics

Division of Neonatal-Perinatal Medicine

University of North Carolina at Chapel Hill

Reviewer 2 Feedback:

1) Introduce a comparative analysis where possible, or discuss the limitations of the study design regarding the absence of a control group. This could help contextualize the findings within the broader literature on neonatal resuscitation practices.

Response:

Our access to a control group would be historically remote (using data from a former newborn resuscitation trial in these facilities conducted a few years prior) or retrospective (using medical record abstraction with little data on actual resuscitation care). As such, in our judgement, the potential comparators are not robust enough to draw conclusions. To address the lack of a comparator group, we added a sentence to the limitations section of the discussion:

“The study design did not include comparative analysis, which limits our understanding of the effect of establishing the registry.” (lines 349-350)

Of note, the third paragraph of the discussion section includes a comparison of our findings to our prior study in the DRC as well as prior literature from Tanzania and Uganda (lines 265-268).

2) Expand the discussion on the implications of registry findings for neonatal care practices and policy, both within the DRC and globally. Highlighting specific variables that were not collected but are critical for improving neonatal outcomes could be particularly insightful.

Response:

We expanded the discussion on implications of registry findings with attention to our findings of excessive suctioning and interruptions in PPV (lines 268-269), the subjectivity of evaluation of breathing (lines 300-302) and the need to expand the target population for evaluating quality resuscitation (lines 322-326). We also highlighted the lack of intrapartum data currently collected in the registry and the importance of addressing the intrapartum period to improve neonatal outcomes (lines 343-344).

3) Streamline the introductory and methodology sections to maintain a sharper focus on the study's objectives, findings, and implications. Reducing descriptive content that does not directly contribute to understanding the study's impact could enhance readability and engagement.

Response:

We revised the introductory and methodology sections per your suggestion to be more succinct and maintain a sharper focus.

4) Provide a more detailed analysis of the barriers and facilitators encountered during the registry's implementation, including how these challenges were addressed or could be overcome in future research. Sharing lessons learned in this aspect could be extremely valuable to others looking to implement similar registries in low-resource settings.

Response:

We re-organized the results section to cluster our analysis of barriers and facilitators in one paragraph, and bolstered this analysis with more detail regarding lessons learned (lines 183-199). We also added more detail to the discussion section regarding the burden of paper record-keeping (lines 246-249), the challenges around midwives consistently using NeoBeat (lines 352-355) and the resources involved in adding research staff to conduct observations with Liveborn (lines 361-362).

Reviewer 3 comments:

1) It is crucial that feedback is provided to the ILCOR neonatal group so that any revision of the Utstein criteria can be updated whenever appropriate. It is however “the hope of the Neonatal Utstein Working Group that, at minimum, all specified core data elements will be reported using the standardized definitions provided here.” It would be useful to provide feedback not only about what data was difficult to collect, but in what way (if any) the authors planned to explore collecting more of the core data elements. Barriers to collection are crucial to discuss and solutions may need to be creative. If this is not possible, it may be that the key components of this manuscript would be better published as a letter.

Response:

We added more detail regarding challenges with collecting certain data elements to the results section, with a focus on core data elements (lines 183-199; see response to #4 from reviewer 2). We also highlighted the new core variables we have since added to the registry in the discussion (lines 246-253).

---

## [Decision Letter · Decision Letter 2]

26 Feb 2025

PONE-D-24-30580R2Establishment of a neonatal resuscitation registry in the Democratic Republic of the Congo: An open cohort studyPLOS ONE

Dear Dr. Patterson,

Thank you for submitting your manuscript to PLOS ONE. After careful consideration, we feel that it has merit but does not fully meet PLOS ONE’s publication criteria as it currently stands. Therefore, we invite you to submit a revised version of the manuscript that addresses the points raised during the review process.

We look forward to receiving your revised manuscript.

Kind regards,

Sebastian Schnaubelt, MD, PhD

Academic Editor

PLOS ONE

Journal Requirements:

Reviewers' comments:

Reviewer's Responses to Questions

**Comments to the Author**

1. If the authors have adequately addressed your comments raised in a previous round of review and you feel that this manuscript is now acceptable for publication, you may indicate that here to bypass the “Comments to the Author” section, enter your conflict of interest statement in the “Confidential to Editor” section, and submit your "Accept" recommendation.

Reviewer #1: All comments have been addressed

Reviewer #4: All comments have been addressed

2. Is the manuscript technically sound, and do the data support the conclusions?

Reviewer #1: Yes

Reviewer #4: Yes

3. Has the statistical analysis been performed appropriately and rigorously? 

Reviewer #1: N/A

Reviewer #4: Yes

4. Have the authors made all data underlying the findings in their manuscript fully available?

Reviewer #1: Yes

Reviewer #4: Yes

5. Is the manuscript presented in an intelligible fashion and written in standard English?

Reviewer #1: Yes

Reviewer #4: Yes

6. Review Comments to the Author

Reviewer #1: Dear authors,

thank you for addressing our comments. You describe in your manuscript the establishment of a neonatal resuscitation registry in the Democratic Republic of the Congo. You describe the findings you have collected in this first period and compared them to the recommended variables by ILCOR. Furthermore and very interesting you describe the challenges and barriers in LMIC such as DRC and propose additional variables for tailored to these settings.

Overall I want to congratulate you on this great manuscript

Reviewer #4: Thank you very much for your revision - the manuscript has substantially changed. The comments refer to the very last version of the manuscript:

Major points:

Methods

1. Were there any quality checks or inter-rater reliability checks given the diversity of background of raters (midwives, doctors, etc)?

2. How were you handling missing data?

Results

1. Were there any differences in baseline characteristics between observed and non-observed newborns?

2. It might be beneficial to see the differences between two facilities.

Minor points:

Introduction

Lines 65-66: I believe you can describe the seven domains fully, otherwise it looks a bit incomplete.

Results, Table 1

Text describing Table 1 is fragmented (for example, it is separated by description of Figure 3). It would be easier to read the manuscript if all the fragments were put together.

7. PLOS authors have the option to publish the peer review history of their article (what does this mean? ). If published, this will include your full peer review and any attached files.

**Do you want your identity to be public for this peer review?** For information about this choice, including consent withdrawal, please see our Privacy Policy .

Reviewer #1: No

Reviewer #4: No

---

## [Author Response · Author response to Decision Letter 2]

9 Apr 2025

April 9, 2025

Sebastian Schnaubelt, MD, PhD

Academic Editor

PLOS ONE

Dear Dr. Schnaubelt,

Thank you for your continued consideration of our manuscript, “Establishment of a neonatal resuscitation registry in the Democratic Republic of the Congo: An open cohort study” (PONE-D-24-30580). Following is a point-by-point response to the reviewer’s comments to match the revised manuscript we have submitted.

Sincerely,

Jackie K. Patterson, MD, MPH

Associate Professor of Pediatrics

Division of Neonatal-Perinatal Medicine

University of North Carolina at Chapel Hill

Reviewer 4 Feedback:

Methods

1. Were there any quality checks or inter-rater reliability checks given the diversity of background of raters (midwives, doctors, etc)?

Response:

We did not perform quality checks or inter-rater reliability of the observations. We added a sentence explicitly stating this in the limitations of the discussion (lines 377-379).

Of note, in the discussion, we detailed that Liveborn Observation app registrations were likely delayed and that observers with clinical responsibilities may have competing tasks; thus, we began restricting data collection to study nurses only. In the future directions in the last paragraph of the discussion, we mentioned our plans to validate the accuracy of observational data compared to audio-video recording, which is currently underway.

2. How were you handling missing data?

Response:

We queried all variables reported as unknown to try to capture as much data as possible. Responses that remained unknown after querying were treated as missing. We added the following sentences in lines 117-119: “All responses of “unknown” were queried to facilitate optimal data capture. Responses that remained unknown after querying were treated as missing.”

Results

3. Were there any differences in baseline characteristics between observed and non-observed newborns?

Response:

Yes, the observed population was less sick compared to the non-observed newborns. We have added Supplemental Table 1 to detail these differences and briefly summarized the differences in the results text as follows:

“The cohort we observed with the Liveborn app was biased towards a less sick

population with lower rates of Cesarean section (0.3% vs 2.2%), prematurity (12.5% vs

17.9% born at 28-36 weeks gestation), low birth weight (9.3% vs 12.8% birth weight

1500-2499g), APGAR 1min ≤ 5 (4.7% vs 7.4%), and neonatal deaths before discharge

(0.9% vs 2%) compared to those not observed [Supplemental Table 1].”

(Results, Lines 209-213)

4. It might be beneficial to see the differences between the two facilities.

Response:

Facility 2 cares for a sicker population compared to facility 1. We have added Supplemental Table 2 to detail these differences and have briefly summarized these differences in the results text as follows:

“Baseline characteristics by facility are notable for a sicker population at facility 2 with

higher rates of Cesarean section (2.6% vs 0%), birth weight ≤ 1499 grams (4.2% vs

0.4%) and 1500-2499 grams (13.6% vs 8.8%), 28-36 weeks gestation (20.1% vs 11%),

APGAR 1 min ≤ 5 (8.5% vs 3.9%), and stillbirths (2.9% vs 1.1%) along with a lower rate

of neonatal transfer to another facility (0.3% vs 2.1%) [Supplemental Table 2].”

(Results, Lines 214-218)

Of note, when our statistician was generating these two supplemental Tables, he noted a small error that the registry data was pulled from September 1, 2022 until August 30, 2023 instead of until August 31, 2023 (missing one day of data). All data now reflect that additional day of data to include a full year of the registry through August 31, 2023. We have updated the numbers in line 146, line 149, line 150, line 161, line 162, and Figure 2 accordingly. While the numbers changed slightly, the manuscript's conclusions remain the same.

Introduction

5. Lines 65-66: I believe you can describe the seven domains fully, otherwise it looks a bit incomplete.

Response:

We listed out the seven domains fully on lines 62-63.

Results, Table 1

6. Text describing Table 1 is fragmented (for example, it is separated by description of Figure 3). It would be easier to read the manuscript if all the fragments were put together.

Response:

In the published version of the manuscript, this legend will be placed directly beneath Figure 3 so that it does not separate the text describing Table 1.

---

## [Editor Report · Decision Letter 3]

24 Apr 2025

Establishment of a neonatal resuscitation registry in the Democratic Republic of the Congo: An open cohort study

PONE-D-24-30580R3

Dear Dr. Patterson,

We’re pleased to inform you that your manuscript has been judged scientifically suitable for publication and will be formally accepted for publication once it meets all outstanding technical requirements.

Kind regards,

Sebastian Schnaubelt, MD, PhD

Academic Editor

PLOS ONE

---

## [Editor Report · Acceptance letter]

PONE-D-24-30580R3

PLOS ONE

Dear Dr. Patterson,

I'm pleased to inform you that your manuscript has been deemed suitable for publication in PLOS ONE. Congratulations! Your manuscript is now being handed over to our production team.

Kind regards,

on behalf of

Dr. Sebastian Schnaubelt

Academic Editor

PLOS ONE